# The efficacy of T790M mutation testing in liquid biopsy—Real clinic data

**Paweł Krawczyk**[1], **Luiza Grzycka-Kowalczyk**[2], **Justyna Błach**[1,3]*, **Katarzyna Reszka**[4], **Izabela Chmielewska**[1], **Robert Kieszko**[1], **Magdalena Wójcik-Superczyńska**[1], **Michał Szczyrek**[1], **Tomasz Jankowski**[1], **Janusz Milanowski**[1]

1 Department of Pneumonology, Oncology and Allergology, Medical University of Lublin, Lublin, Poland, 2 Department of Radiology and Nuclear Medicine, Independent Public Clinical Hospital no. 4 in Lublin, Lublin, Poland, 3 Department of Clinical Immunology, Medical University of Lublin, Lublin, Poland, 4 Institute of Genetic and Immunology GENIM LCC in Lublin, Lublin, Poland

* justynablach91@gmail.com

**Data Availability Statement:** All relevant data are within the manuscript and its Supporting Information files.

## Abstract

Osimertnib is still widely used in the treatment of NSCLC patients who have previously received erlotinib, gefitinib or afatinib and have developed resistance to these drugs mediated by the T790M mutation in exon 20 of *EGFR* gene. We assessed the results of T790M mutation testing in liquid biopsy by Entrogen test and real-time PCR technique in routine clinical practice. Analysis was conducted in 73 plasma samples from 41 patients with locally advanced or metastatic lung adenocarcinoma treated with first- or second-generation of EGFR TKIs. We detected T790M mutation in 18 patients (43.9% of patients, 24.6% positive tests in 73 samples). The incidence of T790M mutation in liquid biopsy was significantly higher in patients with T3-T4 tumors compared to patients with T0-T2 tumors (p = 0.0368, $\chi^2$ = 4.36). Median PFS at the time of progression according to RECIST was significantly (p = 0.0444) higher in patients with T790M mutation than in patients without this mutation (22.5 vs. 15 months). Our results confirmed that T790M mutation is more often detected in patients with a large tumor spreading in the chest and with the long duration of response to first- or second generation of EGFR TKIs. The low sensitivity of the real-time PCR technique in T790M mutation detection could be partially compensated by repeating the tests.

## Introduction

Osimertinib, the third-generation of EGFR tyrosine kinase inhibitors (TKIs), can be used for the first-line treatment of patients with locally advanced or metastatic NSCLC, with mutations in exon 18–21 of *EGFR* (epidermal growth factor receptor) gene [1]. However, the first registration of osimertinib was much narrower. Osimertinib was registered for treatment of NSCLC patients who have progressed after initially successful therapy with first- or second-generation of EGFR TKIs. In this indication, osimertinib is used in patients with detectable T790M (Thr790Met) mutation in exon 20 of the *EGFR* gene. T790M mutation could be detected in tumor or in lymph node re-biopsy or in free circulating tumor DNA (cfDNA) from liquid biopsy [2]. Currently, osimertinib being more frequently used in first-line

**Funding:** The authors received no specific funding for this work.

**Competing interests:** The authors have declared that no competing interests exist.

treatment, and its use in 2nd or 3rd lines is steadily declining. However, patients treated with older generation of EGFR TKIs are still frequently present in routine clinical practice. Therefore, the T790M mutation examination is still widely used in molecular diagnosis of NSCLC patients.

Selection of a tumor cell clone with the T790M mutation is the most common cause of resistance to first- and second-generation of EGFR TKIs. It occurs in over 50% of patients at the time of progression. Other causes of resistance to EGFR TKIs include amplification of *MET* gene (approximately 20% of patients), small cell or squamous cell lung cancer transformation, loss of the primary *EGFR* gene mutation, secondary mutations in *KRAS* or *PIK3CA* genes, and the occurrence of disorders in *ALK* gene [3].

AURA3 clinical trial showed significant superiority of osimertinib over platinum-based chemotherapy in advanced NSCLC patients with T790M mutation in *EGFR* gene and with progression on prior EGFR TKIs treatment. The median progression-free survival (PFS) was significantly longer in patients who received osimertinib than chemotherapy (10.1 months vs. 4.4 months). Moreover, the objective response rate (ORR) was significantly better in first compared to second group of patients (71% vs 31%) [2]. In 2020, final overall survival (OS) results of AURA3 have been reported. The median OS was 26.8 months versus 22.5 months for patients treated with osimertinib and chemotherapy, respectively. The percentage of patients with 24- and 36-months survival was also estimated, which was 55% versus 43% and 37% versus 30%, respectively. Data on prolonged survival in patients receiving osimertinib were obtained despite the crossover phenomenon [4].

The discussed results of the AURA3 study emphasize the need for correct, fast and accurate diagnosis of the T790M mutation in patients treated with the first- or second-generation of EGFR TKIs. Biopsy of primary or metastatic tumor or lymph nodes is often impossible in these patients. Therefore, the ctDNA examination in liquid biopsy comes to the fore in the diagnosis of such patients. The aim of this study was to present the effectiveness of diagnosis of T790M mutation in peripheral blood in qualification to osimertinib therapy in real clinic.

## Material and methods

T790M mutation analysis was conducted in 73 plasma samples from 41 patients (median age: 67 ± 10.95 years, 26 women and 15 men) with locally advanced or metastatic lung adenocarcinoma treated with first- or second-generation of EGFR TKIs. Progression free survival was calculated on the basis of RECIST 1.1 (Response Evaluation Criteria in Solid Tumors). We calculated PFS twice: firstly at the time of blood collection for the T790M mutation test (clinical progression assessed by the investigator) and secondly at the time of actual radiological progression in computed tomography examination. In addition, we determined the size of the tumor and the presence of lymph nodes and distant metastases on the basis of the 8th Edition of TNM (Tumor, Nodes, Metastases) Classification in Lung Cancer. TNM was determined at the moment of blood collection. The sum of the target measurable lesions was also calculated. Detailed characteristics of the patients are presented in Table 1.

For all patients, 4 ml blood samples were collected to the collection tubes with EDTA (Ethylenediaminetetraacetic Acid) anticoagulant and immediately centrifuged twice for plasma collection. Time from the moment of blood collection toplasma separation was 15 minutes. Plasma samples were stored at –20˚C until used. Circulating free cfDNA was isolated with the use of MagMAX Cell-Free DNA Isolation Kit (Applied Biosystems), which uses magnetic beads Bynabeads MyOne. The plasma samples were lysed with proteinase K. After this step, cfDNA was bound to the beads in proper solution. Then, samples were placed in the DynaMag Magnet. Beads with bound DNA were washed with Wash Solution and 80% ethanol. At the

**Table 1. The frequency of T790M mutation in liquid biopsy in NSCLC patients with various demographic and clinical characteristics.** Older and younger patients as well as those with high and low sum of targeted lesions were divided based on the median of these parameters.

| | | Patients with T790M mutation—number (%) | Patients without T790M mutation—number (%) | Statistic—p (χ²) |
|---|---|---|---|---|
| Gender | Male | 6 (40) | 9 (60) | 0.7024 (0.146) |
| | Female | 12 (46.1) | 14 (53.9) | |
| Age | ≥67 years | 12 (54.5) | 10 (45.5 | **0.1395 (2.183)** |
| | <67 years | 6 (31.6) | 12 (68.4) | |
| Tumor size | T0-T2 | 3 (21.4) | 11 (78.6) | **0.0368 (4.36)** |
| | T3-T4 | 15 (55.55) | 12 (44.45) | |
| Lymph nodes metastases | N0-N1 | 11 (50) | 11 (50) | 0.397 (0.717) |
| | N2-N3 | 7 (36.8) | 12 (63.2) | |
| Distant metastases | M0 | 6 (46.1) | 7 (53.9) | 0.8875 (0.039) |
| | M1 | 12 (42.9) | 16 (57.1) | |
| RECIST | Stable disease | 5 (38.5) | 8 (61.5) | 0.6323 (0.229) |
| | Progression | 13 (46.4) | 15 (53.6) | |
| Sum of target lesions | ≥32 mm | 8 (36.4) | 14 (63.6) | 0.2951 (1.096) |
| | <32 mm | 10 (52.6) | 9 (47.4) | |
| Type of primary *EGFR* gene mutations | Exon 19 deletions | 10 (40) | 15 (60) | 0.4773 (1.479) |
| | L858R substitution | 6 (60%) | 4 (40%) | |
| | Rare mutations | 2 (33.3) | 4 (66.7) | |
| Type of primary *EGFR* gene mutations | Frequent mutations | 16 (45.7) | 19 (64.3) | 0.5772 (0.319) |
| | Rare mutations | 2 (33.3) | 4 (66.7) | |
| Type of EGFR TKIs | Erlotinib | 6 (37.5) | 10 (62.5) | 0.7985 (0.45) |
| | Gefitinib | 3 (50%) | 3 (50) | |
| | Afatinib | 9 (47.4) | 10 (52.6) | |
| Type of EGFR TKIs | First generation | 9 (40.9) | 13 (59.1) | 0.6775 (0.174) |
| | Second generation | 9 (47.4) | 10 (53.6) | |

Abbreviations: T–tumor, N–nodes, M–metastases, RECIST–Response Evaluation Criteria in Solid Tumors, TKIs–tyrosine kinase inhibitors, EGFR–epidermal growth factor receptor.

end, cfDNA were eluted with appropriate solution and samples were placed on the DynaMag Magnet. The supernatant containing purified cfDNA was checked for concentration and purity in UV-Vis spectrophotometer.

T790M mutation in ctDNA extracted from the patients' plasma was performed using the ctEGFR Mutation Detection CE-IVD Kit (Entrogen) on Cobas Z 480 real-time PCR system (Roche Diagnostics) according to the manufacturer's instruction: "ctDNA EGFR Mutation Detection Kit For Real-Time PCR for the detection of EGFR somatic mutations circulating DNA in human plasma" for in vitro diagnostic use version 1.3. This kit contains a primer mixture for simultaneous detection of T790M, deletions in exon 19 and L858R mutations, as well as an endogenous control gene. The endogenous control primers amplify an unrelated gene that is used to determine the condition of reagents and whether the reaction contains sufficient amount of amplifiable DNA. Probe for detection of T790M mutation was labelled by FAM (6-caroxyfluorescein), probe for exon 19 deletions–by CY5 (cyanine 5), probe for detection of L858R –by ROX (rhodamine X), and probe for internal control–by VIC (2′-chloro-7′phenyl-1,4-dichloro-6-carboxy-fluorescein). The following cycles threshold (Ct) values have been adopted to confirm or exclude the presence of examined mutations: target Ct ≤36 and VIC Ct

≥19-≤30 was defined as the presence of mutations, while target Ct ≥36 or absent and VIC Ct ≥19-≤30 defined as mutations not detected, VIC Ct >30 defined as sample with DNA under-loaded. In six re-biopsy materials obtained by EBUS-TBNA procedure, DNA was isolated from formalin-fixed paraffin-embedded (FFPE) cytological specimens (cell-blocks). DNA was extracted using QIAamp DNA FFPE Tissue Kit (Qiagen). Isolation was performed according to the manufacturer's instructions. Concentration and quality of isolated DNA was estimated by spectrophotometry. Mutations of *EGFR* gene were identified using the EntroGen *EGFR* Mutations Analysis Kit (Entrogen) on Cobas Z 480 real-time PCR system (Roche Diagnostics) according to the manufacturer's instruction: "EGFR Mutation Analysis Kit For Real-Time PCR Kit for the detection of EGFR exon 18, 19, 20 & 21 somatic mutations" for in vitro diagnostic use. Probes for detection of the following mutations: T790M, exon 19 deletions, L858R, L861Q, S768I, G719X, exon 20 insertions were labelled by FAM and probe for internal control–by VIC. The following Ct values have been adopted to confirm or exclude the presence of all mutations except G719X: FAM Ct ≤38 and VIC Ct >24-≤32 was defined as the presence of mutations, while FAM Ct ≥38 or absent and VIC Ct >24-≤32 defined as mutations not detected, VIC Ct >32 defined as sample with underloaded DNA. The following Ct values have been adopted to confirm or exclude the presence of G719X mutation: FAM Ct ≤37 and VIC Ct >24-≤29 was defined as the presence of mutation, while FAM Ct ≥37 or absent and VIC Ct >24-≤29 defined as mutations not detected, VIC Ct >29 defined as sample with under-loaded DNA.Results of T790M genotyping were correlated with progression free survival, disease stage and other demographic and clinical features. Pearson's chi-square test was used to compare the characteristics of the patient groups divided according to presence of T790M mutation. The U-Mann Whitney test was used for testing equality of population medians among groups with and without T790M mutation. Kaplan-Meier method was used for the comparison of progression free survival probability between the groups with different primary *EGFR* gene mutations and type of EGFR TKIs. Data was expressed as numbers and percentages (for categorized variable) as well as medians (for continuous variables). These tests were performed with Statistica v. 13.1 (Tibco Software, USA). Survival analysis was performed using the Kaplan-Meier estimation method in MedCalc 15.8 (MedCalc Software, Ostend, Belgium) with calculation of the hazard ratio (HR) and 95% confidence interval (CI). We considered p values below 0.05 to be statistically significant.

Before the investigation, the agreement of Ethical Committee of the Medical University of Lublin was obtained (KE-0254/131/2011). Informed, written consent to perform genetic testing was obtained from each patient. The funders had no role in study design, data collection and analysis, decision to publish, or preparation of the manuscript. Our study was supported and funded by Medical University of Lublin. There are no ethical or legal restrictions to sharing our data publicly.

## Results

We performed a total of 73 tests for the detection of T790M mutation in 73 plasma samples, from 41 patients with adenocarcinoma treated with first- or second-generation of EGFR TKIs. The tests were repeated in some patients in order not to miss detection of T790M mutation in liquid biopsy. We detected this mutation in 18 patients (43.9% of patients, 24.6% positive tests in 73 samples). In the group of patients with the presence of T790M mutation, we conducted a total of 35 tests (one in 6 patients, 2 in 7 patients, 3 in 5 patients). In the group of patients without the T790M mutation, we performed a total of 38 tests (one in 13 patients, 2 in 5 patients, 3 in 5 patients). Repeated mutation testing was conducted until patients did not receive further lines of treatment (chemotherapy in T790M-negative patients) or until further treatment was

impossible due to deterioration in performance status. In 6 patients without T790M mutation, a re-biopsy of tumor or lymph nodes was possible to perform (26.1% of T790M-negative patients). The presence of cancer cells in pathomorphological examination and the absence of the T790M mutation in molecular examination was confirmed in all the re-biopsy materials.

We examined three types of mutations in ctDNA: T790M, exon 19 deletions and L858R substitution. In 3 patients previously diagnosed with exon 19 deletions and in 2 patients previously diagnosed with the L858R substitution, we detected these mutations in liquid biopsy, without detecting the T790M mutation (5 studies in total). This proves that the test was sensitive enough to diagnose mutations in liquid biopsy, and that patients were probably truly devoid of the T790M mutation. On the other hand, all tests that were positive for the T790M mutation also showed the presence of the primary mutations in the *EGFR* gene. Unfortunately, the test was unable to determine the presence of other, rare mutations in ctDNA.

Concentration and purity of cfDNA measured by spectrophotometric method had no effect on the frequency of T790M mutation detection. The incidence of T790M mutation in liquid biopsy was significantly higher in patients with T3-T4 tumors compared to patients with T0-T2 tumors (p = 0.0368, $\chi^2$ = 4.36). We detected T790M mutation insignificantly more often in older patients (over 67 years of age) than in younger patients (p = 0.1395, $\chi^2$ = 2.183). Gender and other clinical features (presence of disease progression, presence of lymph nodes and distant metastases, type of primary *EGFR* gene mutations, type of EGFR TKIs, sum of targeted lesions) had no effect on the frequency of T790M mutation detection in liquid biopsy (Table 1). Moreover, the T790M mutation was not detected in 7 patients in whom the only manifestation of progression was a single distant metastasis.

Median PFS at the time of progression according to RECIST was significantly (p = 0.0444) higher in patients with T790M mutation detected in liquid biopsy than in patients without this mutation (22.5 months vs. 15 months, Fig 1).

In contrast, median PFS assessed at the time of blood collection was only slightly higher (p = 0.0901) in patients with the T790M mutation compared to patients without this mutation (16.5 months vs. 12.5 months). Median age and median sum of target lesions were similar in patients with and without mutation.

Median PFS in adenocarcinoma patients treated with first- and second-generation of EGFR TKIs was 17 months. PFS was not related to the type of EGFR TKIs (16 months in patients receiving erlotinib, 22 months in patients receiving gefitinib and 19 months in patients receiving afatinib, Fig 2). The risk of progression was similar in patients receiving first- and second-generation of EGFR TKIs (16.0 months vs. 19 months, HR = 0.9789. 95% CI: 0.4733–2.0247, p = 0.9541).

Median PFS was slightly higher in patients with deletions in exon 19 (23 months) or L858R substitution in exon 21 (17 months) than in patients with rare mutations (5.5 months, Fig 3). Risk of progression was insignificantly higher in patients with primary common mutations than in patients with primary rare mutations (19 months vs. 5.5 months, HR = 0.2905, 95% CI: 0.07516–1.2229, p = 0.0731).

## Discussion

Although the diagnosis of T790M mutation is a standard procedure in case of progression on first- or second-generation of EGFR TKIs, it still generates a number of difficulties. FFPE material containing tumor cells is unavailable in many patients. Tumor or metastatic lymph nodes are smaller after EGFR TKIs compared to their dimension before treatment. Therefore, liquid biopsy is the only material available for these patients. Unfortunately, in peripheral blood the number of ctDNA molecules could be unique. The sensitivity of classic molecular

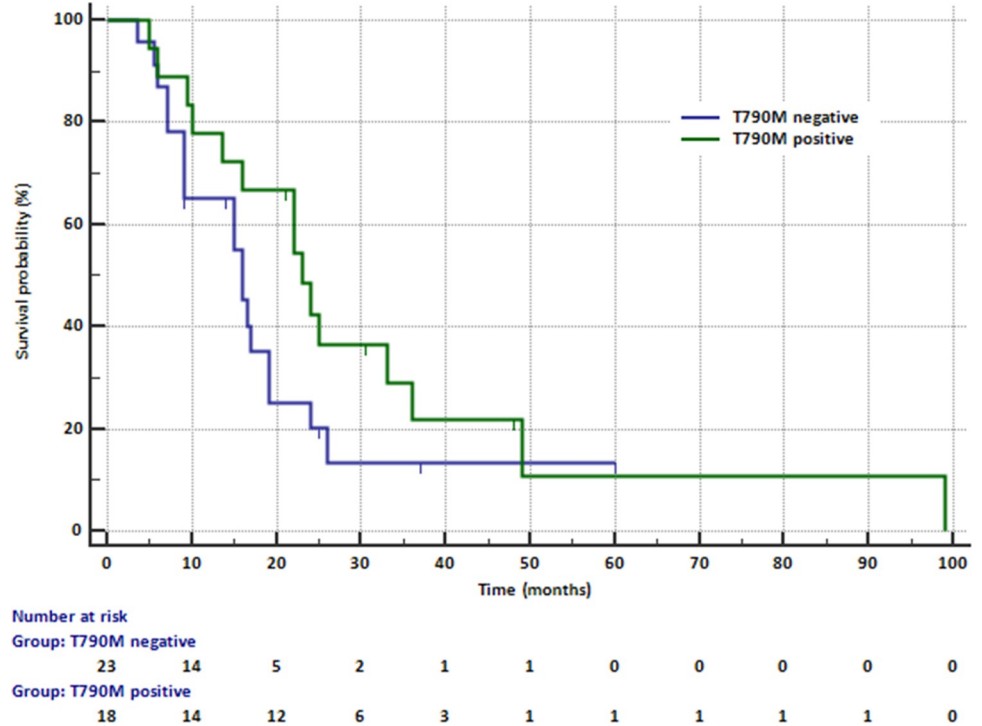

**Fig 1. Median PFS in patients treated with erlotinib, gefitinib or afatinib depending on the possibility of detecting the T90M mutation in liquid biopsy.**

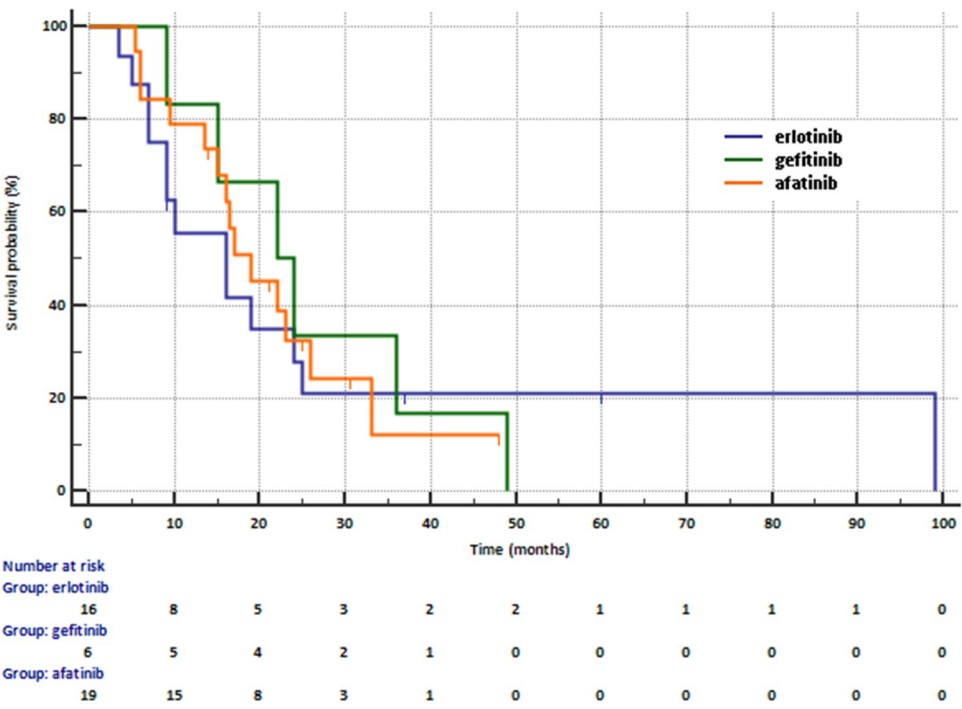

**Fig 2. Progression free survival in patients receiving first-line treatment with erlotinib, gefitinib or afatinib.**

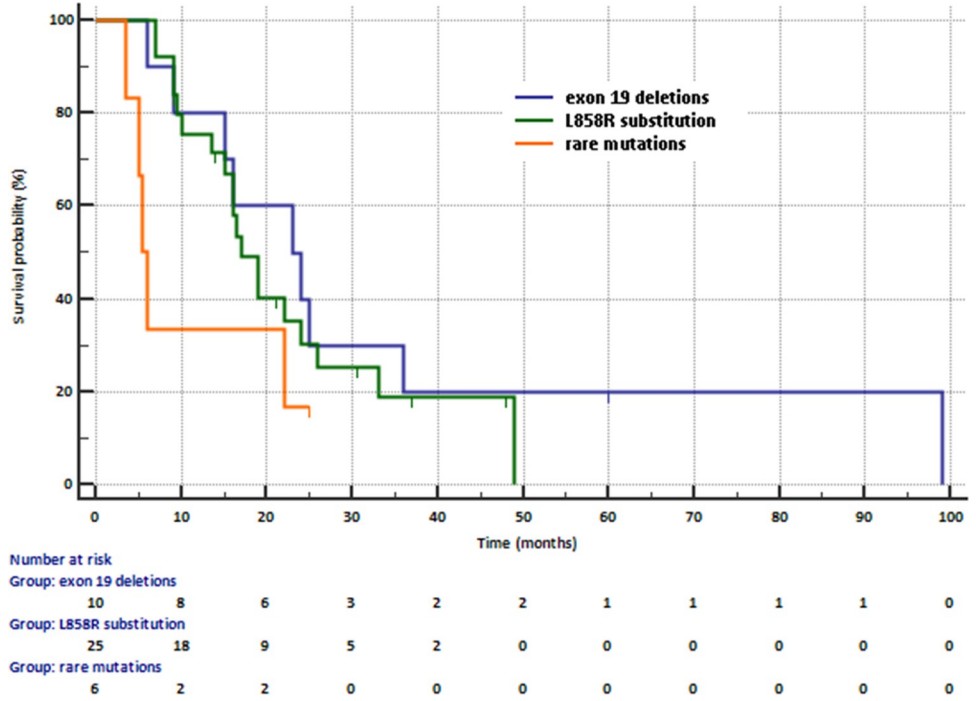

**Fig 3. Progression free survival in patients treated with first- and second-generation of EGFR TKIs according to the type of primary *EGFR* gene mutations.**

methods based on real-time PCR with the use of tests adapted to ctDNA testing is insufficient in many patients. Passiglia et al. indicated that the sensitivity of the Cobas test and real-time PCR techniques in diagnosis of T790M mutations in liquid biopsy did not exceed 70% (30% of false negative results). However, there are reports that classic real-time PCR techniques have even worse sensitivity, which does not exceed 50%. The sensitivity of more advanced PCR technique and next generation sequencing (NGS) exceeds 70% or even 90%, but their availability in many laboratories is limited (Table 2) [5, 6].

The weakness of our study was the inability to determine the sensitivity and specificity of the Entrogen test and real-time PCR technique in diagnosis of T790M mutation in liquid biopsy. Unfortunately, re-biopsy of tumor or lymph nodes was possible only in 6 patients with T790M-negative test in liquid biopsy. The absence of T790M mutation was confirmed in all these materials. Given that T790M mutation was detected in 43.9% of our patients and T790M

**Table 2. Sensitivity and specificity of different genetic methods in T790M mutation diagnosis in liquid biopsy.**

| Methods | Sensitivity | Specificity |
|---|---|---|
| Real-time PCR (Cobas) | 60–64% | 60–98% |
| BEAMing PCR | 70,3–81% | 58–68% |
| ddPCR | 71–87% | 63–100% |
| PNA-LNA PCR | 73–89% | 67–87% |
| NGS | 80–93% | 87–94% |

Abbreviations: PCR–polymerase chain reaction, BEAMing PCR–beads, emulsions, amplification and magnetics PCR, ddPCR–droplet digital PCR, PNA-LNA PCR–peptide nucleic acid-locked nucleic acid PCR, NGS–next generation sequencing

mutation frequency described in literature exceeded 50% of patients, we can conclude that we detected this mutation in almost all truly T790M-positive patients. However, we have achieved this result thanks to several repetitions of the test in some patients.

The ability to detect the T790M mutation does not only depend on the type of methods used for testing. Most researchers believe that the frequency of detecting T790M mutation is higher in patients with advanced stage of the disease. In Li et al. study, T790M mutation rate detected in FFPE tissue and in plasma ctDNA by Cobas tests and in plasma ctDNA by ddPCR method, were 54.5%, 21.3% and 30.4% respectively. T790M positive rate was 52.2% considering all testing methods. The frequency of T790M mutation detected by ddPCR technique significantly rises across stages from IIIB and IVA to IVB (30%, 47.6% and 57.1%, respectively), while no such trend was observed in M1a, M2b and M1c patients [7]. The results of our study confirm these observations. The frequency of T790M mutation detection was significantly higher in patients with large-size tumors infiltrating adjacent structures. It did not depend on the presence of distant metastases.

Most authors agree that T790M mutation is more common in patients with primary exon 19 deletions than in patients with primary L858R substitution. The frequency of T790M mutation detection by various tests ranges from 40% to 73% (55% in pooled analysis of 792 patients from 7 studies) in patients with deletions in exon 19. However, this frequency ranges from 24% to 43% (37% in pooled analysis) in patients with L858R substitution [4]. The Cobas assay with real-time PCR technique were used in the AURA and AURA2 clinical trials to investigate T790M mutation in liquid biopsy and tumor tissue. The mutation was detected in 75% of patients with primary exon 19 deletions, 54% of patients with primary L858R substitution, and only 36% of patients with primary rare *EGFR* gene mutations [8]. Our study group seems to be too small to show statistically significant differences in the frequency of T790M mutation detection in patients with different primary *EGFR* gene mutations. However, a tendency towards a lower incidence of T790M mutation in patients with primary rare *EGFR* gene mutations (33.3%) was also observed in our study.

Study by Goag et al. enrolled 41 NSCLC patients who underwent bronchoscopy to test for T790M mutation. It was identified in 18 (43.9%) patients, and exon 19 deletions were the most significant factor affecting T790M mutation development. The authors identified T790M mutations in 65% of patients with exon 19 deletions, in 21.5% of patients with L858R substitution, and no T790M mutation in patient with rare *EGFR* gene mutations. Moreover, the median time from the start of EGFR TKIs treatment to T790M mutation test was the longest among patients with exon 19 deletions (14.1 months), shorter in patients with L858R or L861Q substitutions (11.3 months) and the shortest in patients with rare *EGFR* gene mutations (2.9 months) [9]. However, Li et al. observed no association between T790M status and duration of first-generation of EGFR TKIs treatment [7]. Our observations are fully consistent with the results of study by Goag et al. The chance of T790M mutation detection increased in patients with long-term treatment with EGFR TKIs. We observed that progression-free survival for EGFR-TKIs was significantly longer in T790M-positive patients than in T790M-neagtive patients. Moreover, the longest PFS was recognized in patients with exon 19 deletions, and the shortest in patients with rare mutations in *EGFR* gene.

Wagener-Ryczek stated that T790M mutation is more common in patients treated with erlotinib or gefitinib (56% of patients) than in patients treated with afatinib (40% of patients). Moreover, this percentage increased in patients treated with EGFR TKIs over 6 months. In such cases, T790M mutation was diagnosed in 64% of patients treated with erlotinib or gefitinib and in 45% of patients treated with afatinib. In patients who received erlotinib or gefitinib, T790M mutation was diagnosed more often in patients with deletions in exon 19 (74% of patients) than in patients with L858R substitution (53%). In contrast, the frequency of T790M

mutation did not depend on the type of primary *EGFR* gene mutation in patients treated with afatinib (44% and 45%, respectively) [10]. In contrast, Jenkins et al. showed no differences in the incidence of T790M mutation in patients treated with erlotinib, gefitinib or afatinib (69.3%, 63.1% and 69.1%, respectively). However, the authors confirmed the observation that T790M mutation is more common in patients with deletions in exon 19 than in patients with L858R substitution only in erlotinib or gefitinib treated group. They found no such differences in patients treated with afatinib [8]. Pereira et al. described that detection of T790M mutation was more likely in patients who were less than 65 years old, with EGFR exon 19 deletions and with duration of first-line treatment of more than 12 months [8]. In contrast to the results of Pereira et al, we have shown that detection of T790M mutation in liquid biopsy was more likely in older NSCLC patients. Our study group seems to be too small to show differences in the frequency of T790M mutation detection in patients treated with erlotinib, gefitinib or afatinib. However, we showed that the median PFS was similar in patients treated with different EGFR TKIs.

Our results and the results of other authors confirm that the detection of T790M mutation depends on many factors. T790M mutation is easier to detect in patients with advanced adenocarcinoma, especially in patients with a large tumor spreading in chest. Another factor contributing to detection of T790M mutation is long duration of response to first- or second-generation of EGFR TKIs. Low sensitivity of real-time PCR technique in T790M mutation detection can be partially compensated by repeating the tests.

## Supporting information

**S1 File.**
(DOCX)

## Author Contributions

**Conceptualization:** Paweł Krawczyk, Justyna Błach.

**Data curation:** Paweł Krawczyk, Luiza Grzycka-Kowalczyk, Justyna Błach, Katarzyna Reszka, Izabela Chmielewska, Robert Kieszko, Magdalena Wójcik-Superczyńska, Michał Szczyrek, Tomasz Jankowski.

**Formal analysis:** Paweł Krawczyk, Luiza Grzycka-Kowalczyk, Justyna Błach.

**Funding acquisition:** Paweł Krawczyk.

**Investigation:** Paweł Krawczyk, Luiza Grzycka-Kowalczyk, Justyna Błach, Katarzyna Reszka, Izabela Chmielewska, Robert Kieszko, Magdalena Wójcik-Superczyńska, Michał Szczyrek, Tomasz Jankowski.

**Methodology:** Paweł Krawczyk, Justyna Błach.

**Project administration:** Paweł Krawczyk.

**Resources:** Paweł Krawczyk, Luiza Grzycka-Kowalczyk, Katarzyna Reszka.

**Supervision:** Paweł Krawczyk, Janusz Milanowski.

**Writing – original draft:** Paweł Krawczyk.

**Writing – review & editing:** Luiza Grzycka-Kowalczyk, Justyna Błach.

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
