## [Decision Letter · Decision Letter 0]

5 Jan 2022

PONE-D-21-32289The efficacy of T790M mutation testing in liquid biopsy – real clinic dataPLOS ONE

Dear Dr. Blach,

Thank you for submitting your manuscript to PLOS ONE. After careful consideration, we feel that it has merit but does not fully meet PLOS ONE’s publication criteria as it currently stands. Therefore, we invite you to submit a revised version of the manuscript that addresses the points raised during the review process.

We look forward to receiving your revised manuscript.

Kind regards,

Hamidreza Montazeri Aliabadi

Academic Editor

PLOS ONE

Journal Requirements:

2. Please ensure that you have specified (1) whether consent was informed, (2) what type you obtained (for instance, written or verbal, and if verbal, how it was documented and witnessed). If your study included minors, state whether you obtained consent from parents or guardians. If the need for consent was waived by the ethics committee and (3) If you are reporting a retrospective study of medical records or archived samples, please ensure that you have discussed whether all data were fully anonymized before you accessed them and/or whether the IRB or ethics committee waived the requirement for informed consent. If patients provided informed written consent to have data from their medical records used in research, please include this information.

The funders had no role in study design, data collection and analysis, decision to publish, or preparation of the manuscript.. 

Reviewers' comments:

Reviewer's Responses to Questions

**Comments to the Author**

1. Is the manuscript technically sound, and do the data support the conclusions?

Reviewer #1: Partly

Reviewer #2: Partly

2. Has the statistical analysis been performed appropriately and rigorously? 

Reviewer #1: I Don't Know

Reviewer #2: Yes

3. Have the authors made all data underlying the findings in their manuscript fully available?

Reviewer #1: No

Reviewer #2: Yes

4. Is the manuscript presented in an intelligible fashion and written in standard English?

Reviewer #1: No

Reviewer #2: No

5. Review Comments to the Author

Reviewer #1: Multiple typos

No new data impacting the clinical practice

Not sure about the value of prognostic analyses as subsequent therapies are not extensively presented

The number of patients is small

There is no correlation with tissue testing

Reviewer #2: The article is written in a correct way. The article is well structured, the number of tables and figures is acceptable and they provide useful information. There are not many bibliographic references (although current). As the authors well comment in the discussion, the work presents some limitations. First, the inability to determine the sensitivity and specificity of the Entrogen test and the real-time PCR technique in diagnosis of T790M mutation in liquid Biopsy. Second, the lack of reproducibility of the assay, as several repetitions are needed to obtain a “valid result”. Third, the lack of contribution of novel results.

I have three questions / comments:

1. Preanalitical steps are critical in the case of liquid biopsies. The authors describe in detail the isolation process of cfDNA and quantification using spectrometry. Have the authors found any association between the concentration of cfDNA obtained and the ability to detect the T790M mutation?

2. The authors comment that the T790M mutation analysis was performed in 73 plasma samples from 41 patients. Could you give us more details of those patients in whom more than one sample was taken? At what point was the second sample taken? Can there be a correlation with the results obtained?

3. In the results section, the authors comment that the test was repeated in a very high percentage of cases (up to 3 replications), justifying that the low sensitivity of the technique for the T790M mutation can be compensated for by repeating the test. Could the authors provide more evidence about this? What criteria were taken into account to decide if a sample should be repeated or not?. The data provided by the authors indicate that the results of the ctEGFR Mutation Detection CE-IVD kit (Entrogen) are not consistent or reproducible, at least for the T790M. Could the authors comment if these results were only obtained for the T790M mutation or also for other mutations detected by the test?

6. PLOS authors have the option to publish the peer review history of their article (what does this mean?). If published, this will include your full peer review and any attached files.

Reviewer #1: No

Reviewer #2: No

---

## [Author Response · Author response to Decision Letter 0]

20 Mar 2022

Response to reviewers

Thank you very much for considering publishing our manuscript in Your journal. Below, we provide a detailed response to all comments from the editor and both reviewers. Corresponding corrections have been made in the revised version of the manuscript.

1. Informed, written consent to perform genetic testing was obtained from each patient. Information about this was added to the text. 

2. The founders had no role in study design, data collection and analysis, decision to publish, or preparation of the manuscript. Information about this was added to the text.

3. Our study was supported from our institution – Medical University of Lublin. We also received funding from our parent institution. Information about this was added to the text.

4. There are no ethical or legal restrictions to sharing our data publicly. Information about this was added to the text.

5. A separate caption for each figure was added in the manuscript.

6. The English language has been corrected accordingly to meet the standard English.

7. Study's minimal data set, as the underlying data used to reach the conclusions drawn in the manuscript, was supplemented in material and methods section. We think that all additional data required to replicate the reported study findings in their entirety have been completed.

8. There was no new data impacting the clinical practice. We are not sure about the value of prognostic analyses as subsequent therapies are not extensively presented. The number of patients is small. There is no correlation with tissue testing.

We would like to thank for valuable comments of the reviewer # 1. We have obtained some new clinically significant results. First of all, the effect of the duration of treatment with 1st and 2nd generation of EGFR TKIs on the chance of developing the T790M mutation is still under discussion. Our voice supported the observation that this mutation is more common in patients with long progression free survival. Secondly, we showed that with multiple repetitions of liquid biopsy, despite the reduced sensitivity of this method, there is a chance of detecting the T790M mutation in almost all patients with this mutation.

Unfortunately, we cannot provide prognostic value for the analysis of the T790M mutation presence. At the time of collecting the material for this article, in Poland there were restrictions on the access to osimertinib (the beginning of the reimbursement of this drug). Therefore, not all patients with detected T790M mutation received osimertinib. Therefore, the overall survival analysis would be unreliable.

We are aware of the limitations of our research. The study group was small, which we commented in the discussion section. It would not be ethical to repeat bronchoscopy in patients who have detected the T790M mutation. In routine clinical practice, liquid biopsy is performed to avoid other invasive methods of specimen collection. Therefore, we limited re-biopsies only to patients who had no T790M mutation and in computed tomography images developed changes available for bronchoscopy.

9. First, the inability to determine the sensitivity and specificity of the Entrogen test and the real-time PCR technique in diagnosis of T790M mutation in liquid biopsy. Second, the lack of reproducibility of the assay, as several repetitions are needed to obtain a “valid result”. Third, the lack of contribution of novel results.

a. Preanalitical steps are critical in the case of liquid biopsies. The authors describe in detail the isolation process of cfDNA and quantification using spectrometry. Have the authors found any association between the concentration of cfDNA obtained and the ability to detect the T790M mutation?

b. The authors comment that the T790M mutation analysis was performed in 73 plasma samples from 41 patients. Could you give us more details of those patients in whom more than one sample was taken? At what point was the second sample taken? Can there be a correlation with the results obtained?

c. In the results section, the authors comment that the test was repeated in a very high percentage of cases (up to 3 replications), justifying that the low sensitivity of the technique for the T790M mutation can be compensated for by repeating the test. Could the authors provide more evidence about this? What criteria were taken into account to decide if a sample should be repeated or not?. The data provided by the authors indicate that the results of the ctEGFR Mutation Detection CE-IVD kit (Entrogen) are not consistent or reproducible, at least for the T790M. Could the authors comment if these results were only obtained for the T790M mutation or also for other mutations detected by the test?

We would like to thank for valuable comments of the reviewer # 2. Concentration and purity of ctDNA measured by spectrophotometric method had no effect on the frequency of T790M mutation detection. We obtained high concentrations of cfDNA from blood plasma, which was sufficient to perform a real-time PCR examination. This is because plasma also contains other, non-cancerous DNA. Our study does not allow for assessment of neoplastic ctDNA at the preanalytical steps. The appropriate sentence has been added to the results section.

Our study presents a real clinic with the date. Repeated mutation testing was conducted until patients did not receive further lines of treatment (chemotherapy in T790M-negative patients) or until further treatment was impossible due to deterioration in performance status. Three repetitions of the tests were most often possible in patients living close to the clinic, who were able to visit the clinic in a short time. A relevant comment has been added to the results section.

The Entrogen test v. 1.3 detects three types of mutations in cfDNA: T790M, exon 19 deletions and L858R. Indeed, in 3 patients previously diagnosed with exon 19 deletions and in 2 patients previously diagnosed with the L858R substitution, we detected these mutations in liquid biopsy, without detecting the T790M mutation (5 studies in total). This proves that the test was sensitive enough to diagnose mutations in the liquid biopsy, and that patients were probably truly devoid of the T790M mutation. On the other hand, all tests that were positive for the T790M mutation also showed the presence of the primary mutations in the EGFR gene. This comment from the reviewer was especially valuable and we have responded to it in the results.

---

## [Decision Letter · Decision Letter 1]

18 Apr 2022

The efficacy of T790M mutation testing in liquid biopsy – real clinic data

PONE-D-21-32289R1

Dear Dr. Blach,

We’re pleased to inform you that your manuscript has been judged scientifically suitable for publication and will be formally accepted for publication once it meets all outstanding technical requirements.

Kind regards,

Hamidreza Montazeri Aliabadi

Academic Editor

PLOS ONE

Additional Editor Comments (optional):

Reviewers' comments:

Reviewer's Responses to Questions

**Comments to the Author**

1. If the authors have adequately addressed your comments raised in a previous round of review and you feel that this manuscript is now acceptable for publication, you may indicate that here to bypass the “Comments to the Author” section, enter your conflict of interest statement in the “Confidential to Editor” section, and submit your "Accept" recommendation.

Reviewer #1: All comments have been addressed

2. Is the manuscript technically sound, and do the data support the conclusions?

Reviewer #1: Yes

3. Has the statistical analysis been performed appropriately and rigorously? 

Reviewer #1: Yes

4. Have the authors made all data underlying the findings in their manuscript fully available?

Reviewer #1: Yes

5. Is the manuscript presented in an intelligible fashion and written in standard English?

Reviewer #1: Yes

6. Review Comments to the Author

Reviewer #1: (No Response)

7. PLOS authors have the option to publish the peer review history of their article (what does this mean?). If published, this will include your full peer review and any attached files.

Reviewer #1: No

---

## [Editor Report · Acceptance letter]

28 Apr 2022

PONE-D-21-32289R1 

The efficacy of T790M mutation testing in liquid biopsy – real clinic data 

Dear Dr. Błach:

I'm pleased to inform you that your manuscript has been deemed suitable for publication in PLOS ONE. Congratulations! Your manuscript is now with our production department. 

Kind regards, 

on behalf of

Dr. Hamidreza Montazeri Aliabadi 

Academic Editor

PLOS ONE